# Structure-aware Knowledge Graph-to-Text Generation with Planning Selection and Similarity Distinction

**Feng Zhao, Hongzhi Zou, Cheng Yan**
Natural Language Processing and Knowledge Graph Lab,
School of Computer Science and Technology,
Huazhong University of Science and Technology, Wuhan, China
{zhaof, hongzhizou, yancheng}@hust.edu.cn

## Abstract

The knowledge graph-to-text (KG-to-text) generation task aims to synthesize coherent and engaging sentences that accurately convey the complex information derived from an input knowledge graph. One of the primary challenges in this task is bridging the gap between the diverse structures of the KG and the target text, while preserving the details of the input KG. To address this, we propose a novel approach that efficiently integrates graph structure-aware modules with pre-trained language models. Unlike conventional techniques, which only consider direct connections between first-order neighbors, our method delves deeper by incorporating Relative Distance Encoding as a bias within the graph structure-aware module. This enables our model to better capture the intricate topology information present in the KG. To further elevate the fidelity of the generated text, Planning Selection and Similarity Distinction are introduced. Our approach filters the most relevant linearized sequences by employing a planning scorer, while simultaneously distinguishing similar input KGs through contrastive learning techniques. Experiments on two datasets demonstrate the superiority of our model.

## 1 Introduction

Knowledge graph (KG) is a structured data representation form that contains rich knowledge information and is more convenient for processes such as information retrieval and reasoning. Although KGs facilitate computational processes, it is difficult for humans to intuitively understand the content in KGs, so the proposed KG-to-text generation task aims to produce correct descriptive text for the input KG. KG-to-text has various applications, like question-and-answer (Pal et al., 2019) and dialogue systems (Zhou et al., 2018). Figure 1 shows a KG and its corresponding text. The main

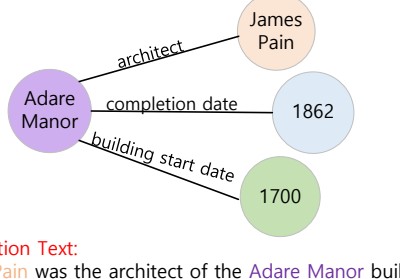

Description Text:
James Pain was the architect of the Adare Manor building which construction started in 1700 and completed in 1862 .

Linearized Sequence:
<H>Adare Manor <R> architect [T] James Pain
<H>Adare Manor <R> building start date [T] 1700
<H>Adare Manor <R> completion date [T] 1862

Figure 1: The KG and its corresponding text. The entities in the text are marked in color. The order of triples in the linearized sequence can control the accuracy of the generated text.

challenges for the task are the **different structures** of the input KG and output texts as well as the need to ensure generated description texts can **completely and accurately** cover the KG information, including all entities and relations.

Benefitting from the powerful capabilities of pre-trained language models (PLMs), methods based on PLMs (Ke et al., 2021) are currently achieving outstanding results on KG-to-text generation tasks. Since there is a structural gap between the graph structure of KGs and the sequence input requirements demanded by PLMs, an important step is to linearize the given KG into an input sequence. The results of many previous works have shown that the final generated texts are significantly influenced by the triple orders of linearized sequences. Many existing works typically use predefined heuristic rules such as breadth-first search (BFS) (Li et al., 2021), or create a content planner using multistep prediction methods to predict of the triple order (Zhao et al., 2020). Each step of these multistep prediction processes may have prediction bias, which results in the final sequence or-

der easily deviating significantly from the ground-truth order. Considering this problem, we train an efficient planning scorer to score the generated results of all linearized sequences corresponding to KG, and pick the sequence with the highest score to be the final input, thus enabling the bias caused by multistep predictions to be avoided.

The linearized sequence may ignore the graph structure information in the corresponding KG, such as the connectivity of nodes and the direction of edges. Some works (Colas et al., 2022; Ke et al., 2021) have improved PLMs by adding graph structure-aware modules to every layer of the transformer encoder. These graph-aware modules only consider the directly connected entities in one triple, i.e., the first-order neighbors in the KG. During the specific text generation process, only the information within each triple is considered, while the connections of the entities between triples are ignored. Therefore, to fully utilize the topological structure information of a KG, we consider the edges in the KG as nodes to compute the shortest path distance among nodes and add it to the weight computation as a bias. Additionally, to enable better distinction of similar input KGs and similar entities in the same KG, we use contrastive learning to enable the transformer encoder to differentiate the input sequences at a finer-grained level. We use two datasets to test our model: WebNLG and DART, and it outperforms the previous models. Our contributions are three-fold:

- We propose an efficient score control-based planning scorer to generate remarkably accurate linearized sequences for text generation.

- Our approach involves capturing and distinguishing the information of a KG using fine-grained control methods, including relative distance-based structure-aware control and contrast learning-based sequence representation distinction.

- Results from two datasets demonstrate the superiority of our model compared to other methods, and our approach has the potential to significantly improve the quality of text generation.

## 2 Related Work

### 2.1 KG-to-Text Generation

To capture the KG structural information, many recent works on KG-to-text generation encode the graph structure directly using graph neural networks (GNNs) (Guo et al., 2019; Zhao et al., 2020; Ribeiro et al., 2020b; Li et al., 2021) or graph-transformers (Schmitt et al., 2020) and then decode into texts. DUALENC (Zhao et al., 2020) feeds the input KG into two GNN encoders for order planning and sentence generation. Graformer (Schmitt et al., 2020) introduces a model that combines relative position information to compute self-attention. Other approaches (Wang et al., 2021b; Liu et al., 2022; Guo et al., 2020; Ribeiro et al., 2020a) first linearize KG into sequences and then feed them into the sequence-to-sequence (Seq2Seq) model for generating desired texts. In this paper, we employ the Seq2Seq model with a planning selector to control linearized sequence orders.

### 2.2 Sequence Order Generation

Existing works (Zhao et al., 2020) have shown that the linearized order of the given triples has an effect on the generated text's quality. Previous works mainly use graph traversal (Li et al., 2021) or multistep prediction (Su et al., 2021; Liu et al., 2022; Zhao et al., 2020) methods for triple order generation. (Li et al., 2021) uses the relation-biased BFS (RBFS) strategy to traverse and linearize KGs into sequences. (Zhao et al., 2020) uses the content planner to select one of the remaining unvisited triples at each step until all triples have been visited. These graph traversal-based approaches are difficult to handle when facing isolated points that are not connected to other points. Methods based on multistep prediction can be incorrect at each step of the chain decisions. Inspired by previous work (Kertkeidkachorn and Takamura, 2020), we use a PLM based efficient planning scorer to control the order generation process.

### 2.3 Pre-trained Language Model

PLMs such as T5 (Raffel et al., 2020) and BART (Lewis et al., 2020) have achieved superior performance on KG-to-text generation tasks. Existing works for solving this task using PLMs fall into two main categories. One category does not change the internal structures of PLMs, but rather optimizes the fine-tuning process by introducing other training tasks. (Li et al., 2021) utilizes the KG reconstruction task and the entity copying task to fine-tune the BART. The other category adjusts the internal structures of PLMs by incorporating

new modules in encoder or decoder. (Ke et al., 2021; Colas et al., 2022) incorporate the graph structure-aware module into the encoder of PLMs. However, they only consider information of directly connected first-order neighbors and ignore the connection information of indirectly connected neighbors. Inspired by (Schmitt et al., 2020), we aggregate the relative distance information into the graph structure-aware module, thus exploiting the information of indirectly connected neighbors.

## 2.4 Sequence Embedding with Contrastive Learning

Due to the power of contrastive learning exhibited in image representation (Radford et al., 2021), many approaches (Gao et al., 2021; Wang et al., 2021a; Reimers and Gurevych, 2019) have applied it to sentence embedding as well. Sentence-BERT (Reimers and Gurevych, 2019) presents a modification of BERT, which employs the Siamese network to derive fixed-sized vectors for input sentences and employs a similarity measure to find semantically similar sentences. SimCSE (Gao et al., 2021) presents the unsupervised methods that apply different hidden dropout masks and the supervised methods that use additional description datasets to generate positives and negatives. Similar to the above sentence embedding approaches, in the KG-to-text generation task we embed linearized sequences of triples using contrastive learning to distinguish similar KGs and similar entities.

## 3 Approaches

Our method not only ensures that the generated text and the target text have the same information, but also enables the model to capture both topological information and specific node content information of the graph for the KG-to-text generation task. To accomplish the first goal, we train a planning scorer that can select the best linearized sequence by scoring all possible input KGs. For the second goal, we first add the graph structure-aware module to the BART encoder. Then, we calculate the cosine similarity of the sequence embeddings from the last layer of the encoder to generate a new loss. Figure 2 represents our model architecture and the following sections describe the contents of these modules.

## 3.1 Problem Formulation

The aim is to generate accurate text to describe the input KG. The input KG consists of some triples and $\mathcal{G} = \{< h, r, t > | h, t \in \mathcal{E}, r \in \mathcal{R}\}$, where $\mathcal{E}$ and $\mathcal{R}$ are sets of entities and relations, respectively. Following (Ke et al., 2021), we linearize the input KG as $\mathcal{G}_{linear} = (w_1, w_2, \cdots, w_m)$, where $m$ is the number of tokens. The target is to generate the text $\mathcal{T} = (t_1, t_2, \cdots, t_n)$, which give an accurate and complete description of the information in the input KG.

## 3.2 Planning Scorer

The purpose of the planning scorer is to compute the scores of all linearized KG sequences to select the highest scoring sequence. Previous methods incorporated GCN-based representation of graphs, but this process does not mainly affect the linearization order of the triples and instead severely affects the training efficiency in the first stage. We therefore propose a planning scorer with a simple structure to further speed up the training efficiency of the model while obtaining the correct linearization sequences. The planning scorer is also divided into two steps: 1) Plan Generation and 2) Plan Evaluation.

### 3.2.1 Plan Generation

We generate all triple orders as needed by Plan generation. Each graph in the WebNLG has no more than seven triples, so we can generate all possible triple orders by exhaustive enumeration. Figure 3 shows a KG with three triples and the six corresponding kinds of linearized sequences, where <H>, <R>, and <T> identify the head entity, relation, and tail entity. For KGs with the number of triples more than seven in the DART dataset, we then randomly sampled only 5,000 possible triplet orders as the result of the planning generation.

### 3.2.2 Plan Evaluation

Planning evaluation performs evaluations for each generated linearized sequence. The structure of the Plan Generation consists of two modules: Linearization Representation and Plan Score.

**Linearization Representation** maps each token in the input linearized sequence into a high-dimensional vector. As shown in Figure 2, we employ BART to achieve the linealization representation of the input sequence. Getting the plan p, we first obtain the linearized sequence $L = (w_1, w_2, \cdots, w_n)$, in which $w_1, w_2, \cdots, w_n$

## 1) Planning Selection

loss ← Plan Socre

bleu

**BART**

Reference:
James Pain was the architect of the Adare Manor building which construction started in 1700 and completed in 1862.

**Planning Scorer**

**FNN**

**Average Pool**

**BART**

Linearized Sequence:
<H>Adare Manor <R> architect <T> James Pain
<H>Adare Manor <R> building start date <T> 1700
<H>Adare Manor <R> completion date <T> 17862

## 2) Text generation

James Pain was the architect of the Adare Manor building which construction started in 1700 and completed in 1862.

**Text generation**

**Decoder**

**Constractive loss**

**Graph-aware Encoder**

Linearized Sequence:
<H>Adare Manor <R> architect <T> James Pain
<H>Adare Manor <R> building start date <T> 1700
<H>Adare Manor <R> completion date <T> 17862

Figure 2: Overview of our model. During the planning selection phase, the planning scorer first generates a linearized sequence of KGs and then computes the resulting score. The BLEU values of the generated texts obtained using references as inputs are used as the labels to train the planning scorer. During the text generation phase, the highest scoring linearized sequences screened by the planning scorer are sequentially fed into the graph structure-aware encoder and the decoder to finish the text generation process.

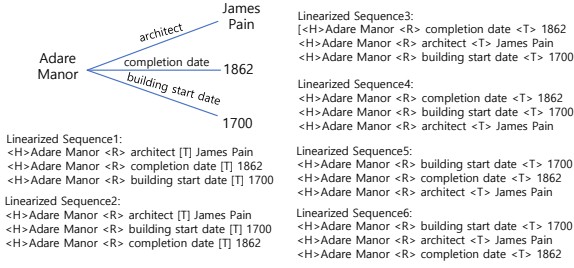

Linearized Sequence3:
[<H>Adare Manor <R> completion date <T> 1862
<H>Adare Manor <R> architect <T> James Pain
<H>Adare Manor <R> building start date <T> 1700

Linearized Sequence4:
<H>Adare Manor <R> completion date <T> 1862
<H>Adare Manor <R> building start date <T> 1700
<H>Adare Manor <R> architect <T> James Pain

Linearized Sequence1:
<H>Adare Manor <R> architect [T] James Pain
<H>Adare Manor <R> completion date [T] 1862
<H>Adare Manor <R> building start date [T] 1700

Linearized Sequence2:
<H>Adare Manor <R> architect [T] James Pain
<H>Adare Manor <R> building start date [T] 1700
<H>Adare Manor <R> completion date [T] 1862

Linearized Sequence5:
<H>Adare Manor <R> building start date <T> 1700
<H>Adare Manor <R> completion date <T> 1862
<H>Adare Manor <R> architect <T> James Pain

Linearized Sequence6:
<H>Adare Manor <R> building start date <T> 1700
<H>Adare Manor <R> architect <T> James Pain
<H>Adare Manor <R> completion date <T> 1862

Figure 3: A KG with three relations and all of its corresponding linearized sequences.

represent each token. Then we feed the linearized sequence L into BART and take the embedding of the decoder's last hidden layer $X = (x_1, x_2, \cdots, x_n)$ as the representation result. Finally the Linearization Representation $C$ is learned using a mean pooling layer by the following equation:

$$C = \frac{1}{n} \sum_{i=0}^{n} x_i, \quad (1)$$

where the linearized sequence has $n$ tokens and $x_i$ denotes the embedding of the i-th token.

**Plan Score** is to estimate the BLEU score based on the Linearization Representation. Here we feed the Linearization Representation $C$ to a feedforward neural network (FNN), and the output is the score:

$$\widehat{y} = C \boldsymbol{W}_{FNN} + b, \quad (2)$$

where $\boldsymbol{W}_{FNN}$ is a linear matrix and $b$ is the bias.

During the training phase, the linearized sequences are fed into BART to generate new texts. And the generated texts and the target texts are used to calculate the BLEU metrics as the target scores $y$. The optimized loss function is as follows:

$$L_{plan} = \sum_{s \in S} \sum_{p \in P(s)} |y_s - \widehat{y}_{s,p}|, \quad (3)$$

where $S$ denotes the corpus and for sample $s$ in the corpus, $P(s)$ denotes the set of generated order plans.

During the testing phase, all possible planning sequences for each sample KG in the dataset are fed into the planning scorer to compute the scores, and the linearized sequences with the highest scores are kept in the file for subsequent generation tasks.

### 3.3 Graph Structure-Aware Model

The linearized sequences are used as the inputs for text generation. We add the graph structure-aware module to each layer of the transformer encoder, and also use the relative distance as a bias when computing the self-attention weights.

**Global Attention**. Each layer in the transformer encoder starts with a Self-Attention module, which captures the global relationships between all tokens of linearized sequences. Assuming that the input linearized sequence is $X$, the

self-attention is computed as follows:

$$\alpha_i = \sigma\left(\frac{X_i \boldsymbol{W}^Q (X\boldsymbol{W}^K)^\top}{\sqrt{d}}\right), \quad (4)$$

$$V_i = \sum_{j=1}^{n} \alpha_{ij}(X_j \boldsymbol{W}^V), \quad (5)$$

where $\boldsymbol{W}^V$, $\boldsymbol{W}^Q$ and $\boldsymbol{W}^K$ are learned parameters. $d$ represents the dimension of word vectors. $\sigma()$ denotes the softmax function.

**Graph Aware Attention**. Global attention captures the global relations between all tokens, but ignores the connectivity among entities and relations in the KG. To further capture the connectivity between entities and relations in the KG, including those not directly connected, we add the structure-aware module that incorporates relative distances behind each Global Attention layer. First the mean pooling is used to generate the representation $X^p \in \mathbb{R}^{m \times d}$ of every component from the output $V \in \mathbb{R}^{n \times d}$ of Global Attention:

$$X^p = pooling(V), \quad (6)$$

where $m$ denotes the sum of the numbers of entities and relations. $n$ is the number of tokens from the linearized sequence. Then we compute the representation $X_l^g$ of each entity and relation by Graph Structure-Aware Attention:

$$\alpha_i^g = \sigma\left(\frac{X_i^p \boldsymbol{W}^{QS}(X^p \boldsymbol{W}^{KS} + q_i \boldsymbol{W}^{KC})^\top}{\sqrt{d}} + \gamma(R)\right), \quad (7)$$

$$\widetilde{X}^g = \sum_{j=1}^{m} \alpha_{ij}^g (X_i^p \boldsymbol{W}^{VS} + q_i \boldsymbol{W}^{VC}), \quad (8)$$

where $\boldsymbol{W}^{QS}$, $\boldsymbol{W}^{KS}$, $\boldsymbol{W}^{KC}$, $\boldsymbol{W}^{VS}$ and $\boldsymbol{W}^{VC}$ are the weight matrices to be learned. $q \in \mathbb{R}^{m \times m}$ denotes the adjacency matrix that records the connectivity among entities and relations, and the value $q$ is either 1 or 0. $\gamma(R) \in \mathbb{R}^{m \times m}$ denotes the relative distance encoding matrix. Following (Schmitt et al., 2020), we consider the relations of KG as graph's nodes and compute the path distance between each pair of nodes to generate the relative distance matrix $R$:

$$R_{ij} = \begin{cases} \infty, & \text{if } \delta(n_i, n_j) = \infty \\ & \text{and } \delta(n_j, n_i) = \infty \\ -\delta(n_j, n_i), & \text{if } \delta(n_i, n_j) > \delta(n_j, n_i) \\ \delta(n_i, n_j), & \text{if } \delta(n_i, n_j) \leq \delta(n_j, n_i) \end{cases}, \quad (9)$$

where $\delta(n_i, n_j)$ denotes the shortest length from $n_i$ to $n_j$. Finally, the graph structure-aware representation $\widetilde{X}^g$ is added to the self-attentive representation $V$ after a residual layer:

$$\widetilde{V} = gather(\widetilde{X}^g) + V, \quad (10)$$

where $gather()$ redistributes the m-dimensional node representation $\widetilde{X}^g$ onto the n-dimensional token representation. Finally, after the encoder-decoder, for the input linearized sequence $G_{linear}$, the text generation loss is formulated as follows:

$$L_T = -\sum_{i=1}^{n} log P(w_j | w_1, \cdots, w_j - 1; G_{linear}), \quad (11)$$

where $P$ is the predicted probability of each token.

### 3.4 Similarity Distinction

For cases of similar input KGs or similar entities in one KG, the generated texts are often confused and erroneous. Therefore we reduce the similarity of different sequence representations by contrastive learning. First we use different sequences in the same batch as simple negative samples for the differentiation of similar input KGs and each linearized KG as its own positive sample. Taking the embeddings $H$ from the encoder's last hidden layer as inputs, we first obtain the representation $r_i$ for each sequence i using a mean pooling layer. We assume that all sequences within a batch are simple negative samples of each other and then compute the contrastive loss by cosine similarity as follows:

$$L_C = -\sum_{i \in P} log\left(\frac{e^{sim(r_i, r_i)/\tau}}{\sum_{j \in J} e^{sim(r_i, r_j)/\tau}}\right), \quad (12)$$

where $P$ denotes the corpus and $J$ denotes the set of training samples in the batch. To distinguish entities in the same KG, we generate some new sequences as hard negative samples for each sequence in a batch. Specifically, we randomly replace each entity in the linearized sequence by other entities with the 15% probability to obtain hard negative samples. All simple negative samples in the same batch and the generated hard negative samples will be used together in the computation of the contrastive loss in Eq. (12).

At last, the total loss $L_{total}$ consists of the text generation loss $L_T$ and the contrastive learning loss $L_C$:

$$L_{total} = L_T + \lambda L_C, \quad (13)$$

where $\lambda$ is the combination coefficient.

## 4 Experiments

### 4.1 Datasets

**WebNLG** (Gardent et al., 2017) is a frequently used dataset. A sample in the dataset contains

Table 1: Results on WebNLG. #Param represents the number of training parameters for models with PLMs. Pre-task represents if models have been pre-trained on additional tasks. ‡ and ♯ indicate that the results are reprinted from (Ke et al., 2021) and (Colas et al., 2022). SOTA-NPT indicates the state-of-the-art (SOTA) model without any pre-training.

| Model | #Param | Pre-task | BLEU | ROUGE | METEOR |
|---|---|---|---|---|---|
| SOTA-NPT♯ (Shimorina and Gardent, 2018) | - | No | 61.00 | 71.00 | 42.00 |
| BART-base♯ (Ribeiro et al., 2020a) | 140M | No | 64.55 | 75.13 | 46.51 |
| T5-base♯ (Ribeiro et al., 2020a) | 220M | No | 64.42 | 74.77 | 46.58 |
| KGPT♯ (Chen et al., 2020) | 177M | Yes | 64.11 | 74.57 | 46.30 |
| JointGT (BART)♯ (Ke et al., 2021) | 160M | Yes | 65.92 | 76.10 | 47.15 |
| GAP‡ - $M^{e,r} + \gamma$ (Colas et al., 2022) | 153M | No | 66.20 | 76.36 | 46.77 |
| Ours | 161M | No | **66.53** | **76.54** | **47.39** |

one to seven triples corresponding to one to five texts. We randomly select one text at a time from one sample as the final target text and replace the underscores with spaces. Following (Ke et al., 2021), we use the version 2.0 and the numbers of KG-text pairs in the training/validation/testing are 34,352/4,316/4,224.

**DART** (Nan et al., 2021) is an open-domain dataset with KGs extracted from Wikipedia tables or incorporated from other datasets like Cleaned E2E (Novikova et al., 2017). The numbers of KG-text pairs in the training/validation/testing are 30,348/2,759/5,097.

### 4.2 Implementation Details

For the Planning Scorer, we employ BART-base for Linearization Representation. In the FNN, we utilize 2 hidden layers and the ReLU activation. The Planning Scorer is trained with the learning rate of 0.001, the batch size of 16, the epoch of 10 and Adam (Kingma and Ba, 2015) to optimize the parameters.

For text generation, the hyperparameters are set as follows: batch size: 100, epoch: 60, learning rate: 2e-5 and optimizer: Adam. The $\tau$ in the contrastive learning is 0.05. The maximum input length of Linearized sequences is 256. The total loss is optimized by Eq. 13 and $\lambda$ is set to 0.5. When generating text in the inference time, the beam search size is 5.

Following previous work, the evaluation metrics for experiments on WebNLG are ME-TEOR (Banerjee and Lavie, 2005), ROUGE-L (Lin, 2004) and BLEU-4 (Papineni et al., 2002). For DART, besides BLEU and METEOR we use two additional metrics, MoverScore (Zhao et al., 2019) and BLEURT (Sellam et al., 2020). With the BLEU metrics produced on the validation sets

Table 2: Results on DART. Seven other models are compared with our model. ‡ indicates that the results are reprinted from (Liu et al., 2022).

| Model | BLEU | METEOR | MoverScore | BLEURT |
|---|---|---|---|---|
| Seq2Seq-Att | 29.66 | 0.27 | 0.31 | -0.13 |
| T5-base | 49.21 | 0.40 | 0.53 | 0.43 |
| T5-large | 50.66 | 0.40 | 0.54 | 0.44 |
| BART-base | 47.11 | 0.38 | 0.51 | 0.37 |
| BART-large | 48.56 | 0.39 | 0.52 | 0.41 |
| JointGT‡ | 54.24 | **0.44** | 0.64 | **0.59** |
| S-OSC‡ | 62.01 | 0.43 | 0.64 | 0.49 |
| Ours | **62.67** | **0.44** | **0.65** | 0.49 |

of both datasets, we select the best performing model.

### 4.3 Main Results

The WebNLG results displayed in Table 1 represent that our model outperforms other models in overall performance including three metrics, i.e., BLEU, METEOR, and ROUGE. Regarding the BLEU metric on WebNLG, our model outperforms the SOTA Seq2Seq model without any pre-training by 5.53%. Compared with BART and T5, our models improve by 1.98% and 2.11%, respectively. Both JointGT and KGPT utilize additional pre-training tasks such as graph-text alignment and text or graph reconstruction. The BLEU improvements of 2.42% over KGPT and 0.61% over JointGT show that our model performs well without additional pre-training tasks. Compared with GAP, which considers only directly connected first-order neighbors in the graph structure-aware module, the improvement of 0.33% on BLEU and 0.62% on METEOR shows that our model better captures the input graph's topological structure.

The results on DART are displayed in Table 2. The large gaps between "Seq2Seq-Att" (Nan et al., 2021) and the other models on all metrics demon-

Table 3: Ablation study for different modules on WebNLG.

| Model | BLEU | METEOR | ROUGE |
|---|---|---|---|
| full model | 66.53 | 47.39 | 76.54 |
| w/o PS | 65.86 | 47.20 | 76.27 |
| w/o RDE | 65.82 | 47.13 | 76.25 |
| w/o DK | 66.17 | 47.41 | 76.29 |

strate the power of PLMs for KG-to-text generation tasks. Compared to the method using only PLMs (e.g. T5-large), our model has an improvement of 12.01% on BLEU, 0.04% on METEOR, 0.11% on MoverScore and 0.05% on BLEURT. The 8.43% improvement achieved in the BLEU metric also shows that our model makes more efficient use of the graph structure. We observe improvements of 0.66% on BLEU, 0.01% on METEOR and 0.01% on MoverScore, over S-OSC. To summarize, our model generates more fluent and accurate texts than the other models.

## 5 Analysis

### 5.1 Ablation Study

In additional experiments on WebNLG, we evaluated the impact of the Planning Scorer, relative distance encoding in the graph structure-aware module and the distinction of KGs on the performance of our model. Besides the full model, we also construct the following three variations: without Planning Scorers (w/o PS) and just relying on orders of the input triples, without relative distance encoding in the graph structure-aware module (w/o RDE) and without distinction of KGs (w/o DK).

The results in Table 3 demonstrate that all three modules impact the performance of our model on BLEU, METEOR and ROUGE metrics. By removing the planning scorer (w/o PS) and the relative distance coding in the graph structure-aware module (w/o RDE), the BLEU metrics drop by 0.67% and 0.71%, respectively, which reflects the importance of the linearized orders and the information abtained from nodes that are not directly connected. Removing the distinction of KGs (w/o DK), the small reductions of BLEU and ROUGE metrics indicate that contrastive learning can improve the accuracy of generated texts to some extent.

### 5.2 Linearized Order of Triples

To confirm our planning scorer's effectiveness, we compare it with the RBFS multistep prediction ap-

Table 4: Results of different methods. PS denotes Planning Scorer. RBFS and RDFS (Li et al., 2021) denote the relation-biased BFS and the relation-biased DFS, respectively. RS denotes the random sorting order.

| Methods | BLEU | METEOR | ROUGE |
|---|---|---|---|
| PS | 66.53 | 47.39 | 76.54 |
| RBFS | 66.30 | 47.28 | 76.34 |
| RDFS | 65.78 | 47.04 | 76.02 |
| RS | 58.22 | 46.17 | 73.09 |

Table 5: BLEU scores for the different models on DART with different graph sizes.

| Model | #Triples | | |
|---|---|---|---|
| | 1-3 | 4-6 | $\geq 7$ |
| JointGT | 75.46 | 56.14 | 47.36 |
| w/o PS | 73.37 | 57.50 | 50.10 |
| full model | 73.98 | 60.44 | 54.30 |

proach and the random sorting order on WebNLG. Table 4 displays the results of different approaches to generate orders of triples. The large drop-off between the random sorting order (RS) and the other methods on the three metrics suggests that linearized orders of triples significantly impact the effectiveness of generated texts. The planning scorer outperforms RBFS and RDFS in three metrics indicating the advantage of evaluating the linearized order of triples from a holistic perspective.

### 5.3 KG Size

To verify the effectiveness of the graph structure-aware module with relative distance encoding, we classify DART into "1-3", "4-6" and "$\geq 7$" according to the number of triples and compare the BLEU metrics for generating texts. Table 5 displays the BLEU metric results of JointGT, our model without the Planning Scorer (w/o PS), and our full model under different conditions. From the top two results, we can observe that in the "1-3" case, JointGT outperforms our model without the Planning Scorer (w/o PS) by 2.09%, which indicates that for small KG inputs, relative distance coding may present redundant information leading to poor results. With the increase of the triples' number, the results of JointGT are gradually overtaken by "w/o PS", which indicates that the relative distance coding can obtain more information about the graph structures of larger graphs with more indirectly connected nodes. The bottom two results show that the role of Planning scorer becomes more important as the number of input triples increases.

Table 6: Examples of the generated texts and linearized KGs produced by JointGT and our model on WebNLG and DART, where **bold** represents entities in KGs.

| | | | |
|---|---|---|---|
| **Real** | Knowledge Graph | 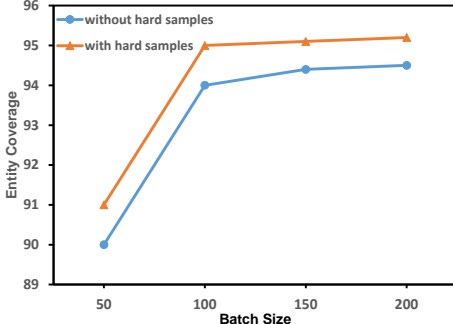 | |
| | Reference | **Musician Andrew White** began in **2003** and is associated with artists **Marry Banilow** and the **Kaiser Chiefs** . He was signed to the **defunct Universal records** . | The **United States**, where the capital is **Washington DC**, includes the ethnic group of **African Americans**. **Auburn** is located in the country and is part of **Lee County in Alabama**. |
| **JointGT** | Linearized KG | ②→①→④→⑤→③ | ①→②→③→⑤→④→⑥ |
| | Generated Text | The musician **Andrew White** started his career in **2003** and is associated with the musical artist **Marry Banilow** . He was once signed to the **defunct Records label** but now plays for **Kaiser Chiefs** . | **Auburn, Alabama** is in **Lee County, Alabama** which is in **Alabama** in the **United States** where **African Americans** are an ethnic group and the capital is **Washington DC**. |
| **Ours** | Linearized KG | ②→①→④→③→⑤ | ⑤→⑥→④→①→②→③ |
| | Generated Text | **Andrew White** started his career in **2003** and is associated with the musical artist **Marry Banilow** and **Kaiser Chiefs**. He was once signed to the now **defunct Universal Records** . | The **United States** where the capital is **Washington DC**, has the ethnic group of **African Americans**. **Auburn** is part of **Lee County** which is in **Alabama** of the **United States**. |

Figure 4: Entity coverage without hard negative samples and with hard negative samples under different batch sizes.

From the WebNLG example we can see that JointGT reverses the position of node 3 and node 5, resulting in incorrect inference results of "once" and "now" in the generated text. With the correct linearization order, our model makes an effective distinction between node 3 and node 4, which have the same relation. On DART, the long entity chains in the input KG lead to many attributive clauses and unclear expressions in the text generated by JointGT. The linearized order generated by our model is not exactly correct, but under the influence of the graph structure-aware module with relative distance encoding and entity distinction, the generated text is able to correctly and concisely describe the input KG.

## 5.4 Entity Coverage

We explore the effect of contrastive learning on differentiating similar KGs and similar entities by calculating the entity coverage in the generated texts over the entities of input KGs. Figure 4 reports the variation curves of entity coverage with and without hard negative samples under different batch sizes. It can be observed that the entity coverage increases with increasing batch size. The hard negative samples are helpful to improve the entity coverage, but the effect becomes smaller as the batch size increases. We set the batch size to 100 because the entity coverage changes very little after the batch size reaches 100.

## 5.5 Qualitative Analysis

Table 6 shows examples of generated texts from our model and JointGT on WebNLG and DART.

## 6 Conclusion

The paper presents a two-step training model for KG-to-text generation including planning scorer and text generation. In the first step the planning scorer first obtains the linearized KG order with the highest score. Then the second step uses a graph structure-aware module with relative distance encoding and KG distinction by contrastive learning to ensure that the model captures the topological information and specific entity features in the input KG. Additional experiments and qualitative analyses indicate our model outperforms the existing KG-to-text approaches on two datasets.

## Limitations

Since the Planning Scorer needs to score all possible linearized sequences of the input KG, the processing becomes less efficient as the number of the input triples increases. Therefore our model is not suitable for handling long text or paragraph generation tasks. Because of the large amount of parameters in PLMs, our model also consumes a large amount of GPU resources when performing comparative learning. Four Tesla V100 GPUs are used to train our model, and the maximum batch size that can be set is 312. Finally, our model can accurately describe entities and relations in the KG, but lacks reasoning power. For example, the triples (A,husband,B) and (B,mother,C) as inputs, it cannot generate "A is the father of C". These above limitations are also the direction of our continuing research in the future.

## Acknowledgements

This work was supported in part by National Natural Science Foundation of China under Grants No.62072203.

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
