# OpenReview forum: "Structure-aware Knowledge Graph-to-text Generation with Planning Selection and Similarity Distinction"
_EMNLP/2023/Conference — EMNLP 2023 Main_

### Official Review · Reviewer_uUx9 · 2023-07-28

**Soundness:** 4

**Excitement:**

4: Strong: This paper deepens the understanding of some phenomenon or lowers the barriers to an existing research direction.

**Paper Topic And Main Contributions:**

The paper presents a novel approach to KG-to-text generation that combines graph-structure-aware modules with pretrained language models. It contains a new learnable approach (called planner) to preprocess an input KG for feeding it into a pretrained language model.

**Questions For The Authors:**

- section 3.2.2 : For training the plan evaluator, do I understand correctly that both BARTs in Fig. 2 (both violet and green boxed ones) are kept static and that only the FNN is trained during that stage?
- And does the violet BART in Fig. 2 receive the candidate linearized sequence as input to generate some text, or is it the reference? The figure and also the text (L.277) suggest the latter, but it doesn't make sense to me. The planning scorer is about predicting the BLEU score the generated text from BART would receive if the candidate linearized sequence was its input, no? So shouldn't the linearized sequence be the input for both of them? (obviously, the reference is needed to compute the "gold" BLEU score, but my confusion is about what is fed to the neural models)
- L330: If q denotes the adjacency matrix, should it be $q\in \{0,1\}^{m\times m}$? I mean, is q any real value or only either 1 or 0, depending on whether there is a connection or not? If it can be something else than 1/0, is it a learned parameter?
- L343: Could you explain the meaning of gather in equation 10? Is this to redistribute the m\times d graph representations X onto the n\times d token representations V?
- L386: "one to five texts" Do you mean "one to five sentences"? As far as I know, WebNLG only has one reference text per input KG, no?
- Given that WebNLG has very few disconnected nodes in the input KGs (as far as I know), why did you not evaluate on some other dataset like AGENDA? There, the positive effects of your relative distance encoding in the graph structure-aware module might have been even larger.
- I don't quite understand the meaning of "linearized KG" in table 6. It shows the order of entities. Fig. 1, however, suggests that the linearized sequence is rather determined by the order of triples, i.e., entities need to be repeated if necessary. In table 6, it looks like every entity occurs only once in a linearized sequence. Could you clarify?

**Reasons To Accept:**

- the paper is overall well written and clearly understandable
- the presented model is well motivated and novel
- combining the writing capabilities of pretrained LMs with graph-aware encoders is an important and exciting research direction
- convincing experimental results, including an ablation study showing the benefits of the new model's different components
- stand-alone evaluation of the input preprocessing model (planner), and a study on entity coverage of the generated texts

**Reasons To Reject:**

No big reasons to be found. If you want to be picky, you could highlight the rather small effects shown in the ablation study, which puts a little doubt on the importance of the new modeling architecture. More experiments, e.g., averaging the results for different random seeds, might mitigate these concerns, showing the robustness and consistently improved performance of the new model in a more convincing manner but I would not see this as a major reason to reject.

**Reproducibility:**

4: Could mostly reproduce the results, but there may be some variation because of sample variance or minor variations in their interpretation of the protocol or method.

**Reviewer Confidence:**

5: Positive that my evaluation is correct. I read the paper very carefully and I am very familiar with related work.

**Typos Grammar Style And Presentation Improvements:**

- Table 4 is missing the dataset (WebNLG) used for evaluation in the caption (would make it easier to read).
- Previous studies using pretrained language models for KG-to-text generation identify hallucinations as a big problem. PLMs tend to rather parrot back facts from their pretraining texts than stay truthful to the input KG. It would be interesting to see how your improved model fares on the challenge of hallucinations. Especially, the impact of contrastive learning would be interesting to investigate.

---

> ### Author Rebuttal · Authors · 2023-08-29
>
> ### Response to Reviewer uUx9
>
> Thank you for the detailed and constructive comments. We address the questions and concerns in the following remarks.
>
> **1. section 3.2.2 : For training the plan evaluator, do I understand correctly that both BARTs in Fig. 2 (both violet and green boxed ones) are kept static and that only the FNN is trained during that stage?**
>
> Indeed, during the training stage of the planning scorer, we maintain the BART parameters as fixed and solely train the FNN component. This approach aims to produce a planning scorer based on BART that aligns consistently with the model employed for subsequent text generation.
>
> **2. And does the violet BART in Fig. 2 receive the candidate linearized sequence as input to generate some text, or is it the reference?**
>
> Thank you for your careful review. In Figure 2, the candidate linearization sequence is inputted into both BART models. The violet BART model utilizes the reference during the computation of BLEU. We have made revisions to clarify the confusing aspect of Figure 2.
>
> **3. Is q any real value or only either 1 or 0, depending on whether there is a connection or not?**
>
> Indeed, q denotes the adjacency matrix, serving the purpose of discerning the interconnectivity among nodes. A value of 1 in q indicates a connection between nodes, while a value of 0 signifies no connection. Further elaboration on the particulars of this matrix will be provided in the manuscript.
>
> **4. Could you explain the meaning of gather in equation 10?**
>
> As you mentioned, the gather() operation in equation 10 redistributes the m-dimensional node representation X onto the n-dimensional token representation. Equation 10 illustrates the expansion of the m-dimensional node representation to an n-dimensional token representation, which is subsequently processed in the residual layer to derive the final token representation.  To provide a comprehensive understanding of this process, we will add descriptions in the manuscript.
>
> **5. "one to five texts" Do you mean "one to five sentences"? As far as I know, WebNLG only has one reference text per input KG, no?**
>
> The WebNLG dataset we used is from Ke et al., 2021[1] and may have version differences. Each of these knowledge graphs contains one to five references. We randomly select one of them as the target reference during preprocessing.
>
> **6. Given that WebNLG has very few disconnected nodes in the input KGs (as far as I know), why did you not evaluate on some other dataset like AGENDA?**
>
> The entities within the AGENDA knowledge graph exhibit limited connectivity, resulting in the majority of computed relative distance encoding values approaching infinity. Consequently, the relative position encoding function fails to effectively operate. Additionally, when dealing with large knowledge graphs, the planning scorer encounters inefficiencies in enumerating all potential linearized sequences, thereby hindering the training process. Instead of using the AGENDA dataset, we conducted experiments on the DART dataset with a relatively large knowledge graph and achieved good results.
>
> **7. I don't quite understand the meaning of "linearized KG" in table 6.**
>
> The linearized KG is determined based on the ordering of the triples, and the specific results of this linearization are presented in Figure 1. In Table 6, we employ abbreviations to succinctly represent the linearized KG, with the aim of visually demonstrating the ordered sequence of entities.
>
>
>
> [1] Pei Ke, Haozhe Ji, Yu Ran, Xin Cui, Liwei Wang, Linfeng Song, Xiaoyan Zhu, and Minlie Huang. 2021. JointGT: Graph-Text Joint Representation Learning for Text Generation from Knowledge Graphs. In Findings of the Association for Computational Linguistics: ACL/IJCNLP 2021, Online Event, August 1-6, 2021, volume ACL/IJCNLP 2021 of Findings of ACL, pages 2526-2538.

---

### Official Review · Reviewer_wGZN · 2023-08-02

**Soundness:** 4

**Excitement:**

3: Ambivalent: It has merits (e.g., it reports state-of-the-art results, the idea is nice), but there are key weaknesses (e.g., it describes incremental work), and it can significantly benefit from another round of revision. However, I won't object to accepting it if my co-reviewers champion it.

**Missing References:**

- Structural Adapters in Pretrained Language Models for AMR-to-Text Generation (Ribeiro et al., EMNLP 2021)
- Step-by-Step: Separating Planning from Realization in Neural Data-to-Text Generation (Moryossef et al., NAACL 2019)

**Paper Topic And Main Contributions:**

This paper proposes an encoding mechanism for encoding the structure of the input graph for the task of graph(KG)-to-text generation. This approach encodes the distances between nodes in the graph using a relative encoding bias.

Besides the graph encoding, the paper proposes a planning scorer with the goal of better linearizing the KG triples for generation and a contrastive learning loss which uses as negative examples linearized triples with replaced entities.

The method is evaluated on two KG-to-text benchmarks (Weblog and DART) and the presented results show strong performance (measured by automatic text generation metrics) compared to previous approaches. Ablation studies show that each proposed component (planning scores, relative distance encoding and contrastive learning) helps the final performance even though the gains are relatively small.

**Questions For The Authors:**

- Can the authors clarify what are the differences between their approach and Schmitt et al., 2020 in the graph encoding using biases? My interpretation is the the both methods (the method proposed in the paper and Schmitt et al., 2020 are the same in spirit) since they use biases to compute path distances between nodes in the graph.

**Reasons To Accept:**

- The proposed model achieved strong performance on two KG-to-text datasets, as measured by automatic metrics.
- The results indicate that the contrastive learning approach that uses negative entities is effective on entity coverage.

**Reasons To Reject:**

- Since the reported numbers for the automatic metrics are close to previous approaches, it would be very valuable to execute human evaluation in order to validate the improvements that the model made. Automatic metrics are not always reliable, as they can be fooled by certain techniques. Human evaluation can provide a more accurate assessment of the model's performance. It can also help to identify areas where the model could be improved.

- The proposed graph aware attention is very similar to the proposed approach by Schmitt et al. 2020 which reduces the novelty of the approach (please see questions for the authors). Moreover, my current understanding is that the text generation model is not built on top of pretrained LM, which is surprising to me since all previous state-of-the-art approaches are based using some flavor of PLM.

**Reproducibility:**

3: Could reproduce the results with some difficulty. The settings of parameters are underspecified or subjectively determined; the training/evaluation data are not widely available.

**Reviewer Confidence:**

4: Quite sure. I tried to check the important points carefully. It's unlikely, though conceivable, that I missed something that should affect my ratings.

---

> ### Author Rebuttal · Authors · 2023-08-29
>
> ### Response to Reviewer wGZN
>
> Thank you for the detailed and constructive comments. We address the questions and concerns in the following remarks.
>
> **1. Human evaluation can provide a more accurate assessment of the model's performance. It can also help to identify areas where the model could be improved.**
>
> We have added human evaluation to the latest version of the paper. We conducted the human evaluation on WebNLG to further evaluate the generated text. Following Chen et al., 2019[1], we adopt human evaluation criteria such as Factual correctness including Supp (counting the number of facts that co-exist in the KG and generated text) and Cont (counting the facts in the generated texts missing from or contradicting with KG), Language naturalness including NF (evaluating the accuracy and fluency of generated sentences). We randomly selected 100 knowledge graphs for human evaluation. Five native English speakers volunteered to score all the 100 knowledge graphs. The following are the results.
>
> | Methods                      | Supp. | Cont. | NF.  |
> | ---------------------------- | ----- | ----- | ---- |
> | Ground_truth                 | 3.94  | 0.13  | 4.85 |
> | JointGT (Ke et al., 2021[3]) | 3.36  | 0.26  | 4.04 |
> | Our model                    | 3.58  | 0.20  | 4.28 |
>
> The results show that the generated texts of our model are more authentic and consistent with KG than JointGT.
>
> **2. Moreover, my current understanding is that the text generation model is not built on top of pretrained LM, which is surprising to me since all previous state-of-the-art approaches are based using some flavor of PLM.**
>
> Certainly, state-of-the-art models currently rely on the formidable capabilities of PLMs, and our generation model follows suit in this regard. However, there is a generation gap between the graph structure and the sequence structure required by the PLM in KG-to-text tasks. To address this disparity, we employ a BART model that incorporates graph structure-aware modules within the Encoder for text generation. As a result, the PLM-based model is utilized during both training stages, thereby ensures the consistency of order prediction.
>
> **3. Can the authors clarify what are the differences between their approach and Schmitt et al., 2020 in the graph encoding using biases?**
>
> Firstly, our relative distance coding is grounded in the node level, thereby taking into consideration the interconnectivity between nodes and prioritizing the preservation of graph structure-awareness in the generated content. This approach ensures comprehensive coverage and comprehension of KG content in the generated text. In contrast,  the approach proposed by Schmitt et al., 2020[2] encodes at token level, with a focus on the accuracy of the generated text.
>
> Secondly, we have streamlined the process of relative distance encoding, resulting in the elimination of the computation process between tokens within a node.
>
> Finally, we have incorporated relative distance coding into the graph structure-aware module. To the best of our knowledge, no work has been done this way. The results presented  in Table 3 demonstrate the effectiveness of our relative distance coding in improving the performance of the graph structure-aware model.
>
>
>
> [1] Zhiyu Chen, Harini Eavani, Wenhu Chen, Yinyin Liu, and William Y ang Wang. 2019. Few-shot nlg with pre-trained language model. arXiv preprint arXiv:1904.09521.
>
> [2] Martin Schmitt, Leonardo F. R. Ribeiro, Philipp Dufter, Iryna Gurevych, and Hinrich Schütze. 2020. Modeling Graph Structure via Relative Position for Better Text Generation from Knowledge Graphs. CoRR, abs/2006.09242.
>
> [3] Pei Ke, Haozhe Ji, Yu Ran, Xin Cui, Liwei Wang, Linfeng Song, Xiaoyan Zhu, and Minlie Huang. 2021. JointGT: Graph-Text Joint Representation Learning for Text Generation from Knowledge Graphs. In Findings of the Association for Computational Linguistics: ACL/IJCNLP 2021, Online Event, August 1-6, 2021, volume ACL/IJCNLP 2021 of Findings of ACL, pages 2526-2538.

---

### Official Review · Reviewer_5CDR · 2023-08-05

**Soundness:** 3

**Excitement:**

3: Ambivalent: It has merits (e.g., it reports state-of-the-art results, the idea is nice), but there are key weaknesses (e.g., it describes incremental work), and it can significantly benefit from another round of revision. However, I won't object to accepting it if my co-reviewers champion it.

**Paper Topic And Main Contributions:**

This paper focus on the KG-to-Text task. It proposes structure-aware KG-to-Text generation model with planning selection and similarity distinction. It designs Planning Scorer module to predict the sequence of knowledge triples. It integrates graph structure-aware representation in to KG representation in the graph structure-aware encoder. It also use contrastive loss to distinguish the similar KGs.

**Questions For The Authors:**

1. How do you ensure that the order of the  planning scorer predictions is appropriate for the latter
2. What the mean of gather(g) in Eq. (10)?
3. It is better to analysis the effect  of lamda in model performance.
4. One target of KG-to-Text task is diversity. Do you consider this in your experiments?

**Reasons To Accept:**

The experiments demonstrate that the proposed model is slightly better than the baselines.

**Reasons To Reject:**

1. This paper used the BART as generation model to train the Planning Scorer while the text generation is another different model (Graph Encoder-Decoder). It is difficult to ensure that the order of the former predictions is appropriate for the latter. There are inconsistency in this process.
2. There are some contraditions in Eq. (9). The first question is how do we initialize the R_ij? The second question is that whether R_ij is equal to R_ji in the relative distance matrix R. According to the Eq. (9), the computation process must be iterative until it is unchanged. If not, the node id will influence the final distance. Based on this, the distance of R_ij will be equal to R_ji since there is the minus sign for the second case in Eq. (9). This means that R_ij and R_ji are equal in R. It makes the complex computation in Eq. (9) meanless.
3. Some key points are missed, eg. the lamda in Eq. (13). It should analysis the effect of the value of lamda through more experiment.
4. The scale of batch size in Figure 4 seems incorrect.

**Reproducibility:**

3: Could reproduce the results with some difficulty. The settings of parameters are underspecified or subjectively determined; the training/evaluation data are not widely available.

**Reviewer Confidence:**

4: Quite sure. I tried to check the important points carefully. It's unlikely, though conceivable, that I missed something that should affect my ratings.

---

> ### Author Rebuttal · Authors · 2023-08-29
>
> ### Response to Reviewer 5CDR
>
> Thanks for the detailed and constructive comments. We address the questions and concerns in the following remarks.
>
> **1. There are some contradictions in Eq. (9). The first question is how do we initialize the $R_{ij}$? The second question is that whether $R_{ij}$ is equal to $R_{ji}$ in the relative distance matrix R.**
>
> The relative distance matrix, denoted as $R$, serves as a bias in the computation of the graph structure-aware module, enabling the acquisition of information pertaining to nodes that lack direct connections. $R_{ij}$ is generated during the dataset preprocessing stage, based on the relative distances between nodes in the KG. Given the deterministic nature of the distances between nodes in the KG, the initial computation of $R_{ij}$ is non-iterative and conducted in its entirety. Notably, the relative distance calculation is predicated on a directed graph, with the minus sign indicating directionality. $R_{ij}$ takes the minimum of the distance from i to j and j to i. The minus represents the direction, so $R_{ij}$ and $R_{ji}$ are different.
>
> **2. The scale of batch size in Figure 4 seems incorrect.**
>
> To distinguish those similar KGs and those similar entities, we employ a contrastive learning approach. Each linearized KG and itself serve as positive samples for each other. Different linearized KGs in the same batch serve as negative samples to differentiate similar KGs. In order to verify the effect of the batch size on model's effectiveness, we conducted experiments on four scales. To facilitate readability, we have transformed the experimental results in the figure into a tabular format, as shown in the table below. The table displays the entity coverage under varying batch sizes, distinguishing between results obtained without hard samples (w/o HS) and with hard samples (HS), respectively. The results show that the entity coverage increases with the increasing of the batch size. We have optimized Figure 4 in the latest version of the paper.
>
> | batch size |  w/o HS |  HS |
> | :----------: | :------------------------------------: | :---------------------------------: |
> | 50         | 90                                   | 91                                |
> | 100        | 94                                   | 95                                |
> | 150        | 94.4                                 | 95.1                              |
> | 200        | 94.5                                 | 95.2                              |
>
> **3. How do you ensure that the order of the planning scorer predictions is appropriate for the latter.**
>
> The text generation model remains rooted in BART, with the inclusion of a structure-aware semantic aggregation module within the Encoder.  The incorporation of similar structures serves to uphold order prediction consistency. Notably, the structure-aware semantic aggregation module is dedicated to capturing connectivity features among nodes and does not directly influence order prediction.  Observing the experimental results in Table 3, it becomes evident that the introduction of the Planning Scorer yields improvements in the model's effectiveness.
>
> **4. What the mean of gather(g) in Eq. (10)?**
>
> The gather() in Eq.10 redistributes the m-dimensional node representation X onto the n-dimensional token representation. Eq.10 represents that the m-dimensional node representation is expanded to an n-dimensional token representation, which is then computed in the residual layer to obtain the final representation of the token. We will add descriptions in the manuscript.
>
> **5. It is better to analysis the effect of lamda in model performance.**
>
> Keeping the batch size 100 and other parameters unchanged, we conducted experiments on the WebNLG dataset for different values of lamda. The results are tabulated below.
>
> | lamda |   0   |  0.1  |  0.3  |  0.5  |   1   |   2   |
> |-------|:-----:|:-----:|:-----:|:-----:|:-----:|:-----:|
> | **BLEU**  | 66.17 | 66.32 | 66.49 | 66.53 | 66.45 | 66.44 |
>
> The results show that the BLEU value is maximum when the lamda size is 0.5. Whereas, the model is least effective when lamda is 0, i.e., with no distinction of KG (w/o DK), which indicates that the DK module is effective. Therefore, we set lamda to 0.5 in our previous experiments.
>
> **6. One target of KG-to-Text task is diversity. Do you consider this in your experiments?**
>
> Yes, the results generated by our model exhibit greater diversity. The diversity of KG-to-text refers to the ability to generate more text with diverse content, which includes the ability to generate text that contains richer entities. We therefore demonstrate the diversity capability of our model by validating the increase in the entity coverage of the generated text. The improved entity coverage in Figure 4 indicates that our model can generate texts with more diverse entities. The purpose of incorporating contrastive learning in our model is to enhance the model's capability to differentiate the content of similar KGs, which leads to more precise and more diverse generated texts. Below are two examples of similar KGs in WebNLG and their corresponding output when using (Text1 and Text2) and not using (TextDK1 and TextDK2) the DK(distinction of KGs) module:
>
>
>
> KG1: < Malaysia, leaderName, Abu Zahar Ujang>, < Malaysia, ethnicGroup, Malaysian Malay>, < Malaysia, ethnicGroup, Malaysian Indian>
>
>
>
> Reference1: Abu zahar ujang is a leader in Malaysia, where there are ethnic groups called Malaysian Malay and Malaysian Indian.
>
>
>
> KG2: < Malaysia, capital, Kuala Lumpur>, < Malaysia, ethnicGroup, Malaysian Malay>, < Malaysia, ethnicGroup, Malaysian Indian>
>
>
>
> Reference2: Malaysian Indian and Malaysian malay are ethnic groups of Malaysia, where the capital is kuala Lumpur.
>
>
>
> Text1: The leader of Malaysia is called jusuf kalla and Malaysian Malay are an ethnic group there.
>
>
>
> Text2: Malaysians Malay and Malay Malay ethnic group are both ethnic groups and Kuala Lumpur is the capital of Malasia.
>
>
>
> TextDK1: Abu Zahar Ujang is the leader of Malaysia where Malaysian Malay and Malaysian Indian are ethnic groups.
>
>
>
> TextDK2: Malaysian Malay and Malaysian Indian are ethnic groups in Malaysia where Kuala Lumpur is the capital.
>
>
>
> Based on the results above, it can be observed that without the DK module, the output sometimes fails to capture crucial information and only generates duplicate content when dealing with similar KGs. For instance, Text1 only presents the common information shared between KG1 and KG2 while disregarding the triplet <Malaysia, leaderName, Abu Zahar Ujang>. Additionally, similar entities within a single KG, such as Malaysian Malay and Malaysian Indian in Text2, are not correctly distinguished. After adding the DK module, these problems are solved. Therefore, the contrastive learning approach improves the diversity of generated texts.

---

### Meta-Review · Area_Chair_qHWG · 2023-09-19

**Recommendation:** 4

**Metareview:**

The paper presents a new framework for knowledge graph to text generation (KG-to-text) that combines graph-structures modules with pretrained language models (PLMs). A planner is also proposed to fuse the input knowledge graph to the PLMs. Although concerns are raised about ablation study for contribution of different components and novelty. Overall, the paper is well written with strong empirical results. Incorporating graph-aware encoders with PMLs is an exciting direction and it can be beneficial to a broader community.

---

### Decision · Program_Chairs · 2023-10-07

**Decision:**

Accept-Main

**Comment:**

The paper presents a new framework for knowledge graph to text generation (KG-to-text) that combines graph-structures modules with pretrained language models (PLMs). A planner is also proposed to fuse the input knowledge graph to the PLMs. Although concerns are raised about ablation study for contribution of different components and novelty. Overall, the paper is well written with strong empirical results. Incorporating graph-aware encoders with PMLs is an exciting direction and it can be beneficial to a broader community.